# REBT Integration in Public and Private Outpatient Settings during COVID-19 Pandemic: Perspectives from Indonesia

Surilena [1,*], Alegra Wolter [1], Michael Vincentius [1] and Adela Teresa [2]

1 Department of Psychiatry, Faculty of Medicine and Health Sciences, Atma Jaya Catholic University of Indonesia, Jakarta 14440, Indonesia
2 Faculty of Medicine and Health Sciences, Atma Jaya Catholic University of Indonesia, Jakarta 14440, Indonesia
* Correspondence: surilena@atmajaya.ac.id

**Abstract:** This quasi-experimental study aimed to investigate how rational emotive behavior therapy (REBT) works in Indonesian outpatient settings, especially in the context of the COVID-19 pandemic. This study evaluated depression, anxiety, and sleep among 60 Atma Jaya Hospital patients (public and private) using several assessment tools (PHQ-9, GAD-7, and PSQI). The majority of participants were under 40 (53.3%), female (58.3%), married (56.6%), and working formally (65%), with either high school or diploma/bachelor's level education (86.6%). After six REBT therapy sessions, significant improvements were evident in anxiety, depression, and sleep quality scores ($p < 0.001$). Participants showed improvements in anxiety and depression during the third and sixth sessions; and in sleep quality during the sixth session. These findings suggest that REBT-based interventions are effective in public and private outpatient settings, highlighting the importance of psychotherapy and cross-division collaboration in the Indonesian healthcare system.

**Keywords:** COVID-19; REBT; psychotherapy; Indonesia





## 1. Introduction

In 2019, the World Health Organization declared COVID-19 a global pandemic, with more than 6 million deaths having occurred globally. The WHO recommended various strategies to reduce transmission, including hand washing, masks, and physical distancing [1]. The Special Capital Region of Jakarta implemented large-scale social restrictions (PSBB) as one of the strategies to reduce the spread of SARS-CoV2 [2]. In addition, the "new normal" era was introduced to help people return to their everyday lives with various adjustments to prevent the spread of COVID-19 [1,2].

The negative impact of a global pandemic can further harm aspects of the mental health of individuals with or without mental disorders. With the widespread fear and uncertainty surrounding the virus, people struggled with anxiety, depression, sleep disorders, and stress during the pandemic [3]. The COVID-19 pandemic has led to many irrational thoughts and beliefs, such as the fear of contracting the virus, uncertainty of its impact, and worry about the future. The overall prevalence rates of depression, anxiety, distress, and insomnia were 31.4%, 31.9%, 41.1%, and 37.9%, respectively [4]. This was highlighted in other research focusing on anxiety (67%) as a major source of psychological impact on the general population [5], as well as how self-isolation can increase depressive symptoms (74.3%) and worsen sleep quality (43.0%) [6].

The general objective of REBT intervention is to increase the ability to manage irrational beliefs in order to be rational when dealing with various situations, events, experiences, or problems, while also increasing logical thinking, self-acceptance, sense of comfort, security, and positive behavior [7]. REBT can help individuals to cope with fears by challenging and changing their irrational beliefs and replacing them with rational, more adaptive ones. Changing thinking patterns and beliefs can significantly change feelings

and behavior. A recent study found that REBT was effective in reducing symptoms of anxiety and depression in individuals who were struggling with the impact of COVID-19. The study by Bechara recruited 120 participants experiencing high levels of anxiety and depression due to the pandemic. The participants were randomly assigned to either the REBT intervention or the control group. After 12 weeks of treatment, the REBT group showed significantly reduced anxiety and depression symptoms compared to the control group [8].

For context, psychotherapy is an essential treatment option for individuals with mental health conditions and is available in both public and private outpatient settings. In Indonesia, psychotherapy is typically provided by licensed mental health professionals, such as psychiatrists and clinical psychologists. Public outpatient centers often provide psychotherapy services at a lower cost, or even for free, for individuals without insurance or limited financial resources. However, mental health professionals are not widely available at the public primary care level, and are typically accessible only in big cities, such as Jakarta and Yogyakarta. On the other hand, private outpatient centers typically offer psychotherapy services for a fee. Private hospitals may also have shorter wait times for appointments and provide more privacy. The choice between these settings may depend on factors such as financial resources and personal preferences. Regardless of the settings, psychotherapy should be considered as a treatment option for individuals seeking mental health support [9–12].

Moreover, REBT-based interventions have been used in the Indonesian healthcare setting to navigate the intersections between physical conditions and psychological well-being. A study conducted by Surilena et al. in 2014 showed how REBT-based therapy improved not only mental health conditions, but also adherence in women living with HIV. The study was conducted on 160 women on treatment ($n = 80$) and a control group ($n = 80$), who underwent 8 weeks of intervention. After the intervention, the REBT-based group showed significant improvement in self-reported ART adherence and reduced SRQ-20 mean scores (a measure of mental health) compared to the control group. The study concluded that REBT-based therapy could be effective in improving mental health and ART adherence in women living with HIV [13].

Given the challenging situation with COVID-19, this quasi-experimental study was designed to evaluate the psychological conditions of patients presenting to the BPJS (Public) and Bonaventura (Private) outpatient settings at the Atma Jaya Hospital, where REBT-based psychological interventions were provided. Each individual received six therapy sessions, each lasting 30–45 min. We aimed to obtain an overview of psychological conditions, especially anxiety, depression, and sleep quality, during the COVID-19 pandemic as well as to determine the future role of REBT-based interventions in Indonesia. The results of this study are expected to create further REBT-based interventions and modules to overcome anxiety, depression, and sleep disorders that can be used by patients and students in hospital and university settings to overcome psychological problems.

## 2. Methods

Design: This research utilized a quasi-experimental approach and implemented six REBT sessions as the intervention. The mental health conditions of all participants were evaluated before and after each session. Quasi-experimental studies can be useful in evaluating the effectiveness of psychotherapy interventions, such as REBT, by comparing the outcomes and establishing whether the intervention therapy has a positive impact [14]. Furthermore, the REBT psychotherapy module was available, adapted, and applicable for use in the Indonesian setting [13].

Participants: The study involved individuals who were receiving outpatient care at the Atma Jaya Hospital, either in public or private settings. Participants were selected using the convenience sampling method during the months of May to November 2021, specifically during the lead researcher's practice hours on Mondays (10:00–13:00, private), Tuesdays (07:00–13:00, public), and Fridays (10:00–13:00, private). Inclusion criteria included con-

senting patients aged 20–65 years old, capable of reading and writing. Individuals who had been diagnosed with an organic mental disorder, psychotic disorder, or severe major depressive disorder with suicide ideation/attempts, self-injury, or psychotic symptoms were excluded from the study.

Instrument: The characteristics of all participants were evaluated based on age, sex, education, marital status, and occupation. To assess depression, the study used the Indonesian version of the Patient Health Questionnaire (PHQ-9), which has been validated with a Cronbach's alpha value of 0.718 [15]. The study used the Generalized Anxiety Disorder Scale (GAD-7) for anxiety, with a Cronbach's alpha of 0.876 for the Indonesian version, 100% sensitivity, and 84.4% specificity [16]. Additionally, this study used the Pittsburgh Sleep Quality Index (PSQI) questionnaire, which consists of nine questions, to measure sleep quality. The Indonesian version of the PSQI questionnaire was validated in 2018, with a Cronbach's alpha value of 0.81 [17].

Analysis: A descriptive analysis was conducted to investigate the proportion of participants experiencing anxiety, depression, and sleep quality disorders based on their demographic characteristics. Furthermore, the study utilized repeated measures of ANOVA to evaluate changes in the symptoms of anxiety, depression, and sleep quality among participants in the first, third, and sixth sessions of the REBT-based intervention. The data distribution was analyzed using the Shapiro–Wilk test. The results of the analysis showed that none of the variables were normally distributed. Anxiety scores before intervention (T0) were $W = 0.90$, $p < 0.001$; after three sessions (T1), $W = 0.90$, $p < 0.001$; and after completion of all intervention sessions (T2), $W = 0.86$, $p < 0.001$. Depression scores were also not normally distributed in any of the measurements (T0: $W = 0.86$, $p < 0.001$; T1: $W = 0.86$, $p < 0.001$; and T2: $W = 0.83$, $p < 0.001$). Only the sleep quality data before intervention were normally distributed (T0: $W = 0.97$, $p < 0.20$). Sleep quality data in other measurements were not normally distributed (T2: $W = 0.71$, $p < 0.001$; T3: $W = 0.85$, $p < 0.001$). Therefore, a further analysis was conducted using a non-parametric approach. Medians and ranges of the values were calculated for anxiety, depression, and sleep quality scores. A repeated measures ANOVA analysis [18] with the non-parametric Friedman test was performed to assess changes in anxiety, depression, and sleep quality symptoms of the participants before the first session, as well as after the third and sixth sessions of REBT-based intervention.

## 3. Results

There was no sample drop-out among the 60 participants in this study. The majority of participants were under 40 (53.3%), female (58.3%), married (56.6%), and working formally (65%), with either high school or diploma/bachelor's education levels (86.6%).

### 3.1. Distribution of Anxiety, Depression, and Sleep Quality by Sex and REBT Sessions

As shown in Table 1, women reported higher rates of anxiety, depression, and poor sleep quality than men. Among female participants who had anxiety in the first session of REBT, 66.67% experienced severe anxiety, 54.55% had moderate anxiety, and 50% had mild anxiety. In the third REBT session, there was an improvement in moderate anxiety (66.67%) and mild anxiety (46.15%). By the sixth session of REBT, mild anxiety had improved in 57.14% of participants, and moderate anxiety had improved in 50%.

For female participants experiencing depression in the first REBT session, moderate depression was present in 80%, severe depression in 76.9%, and mild depression in 54.17%. In the third REBT session, there were improvements in moderate depression (73.3%), mild depression (61.54%), and severe depression (46.67%). The depression continued to improve in the sixth REBT session, with mild depression improving in 68.14% of participants and moderate depression improving in 40%.

**Table 1.** Distribution of anxiety, depression, and sleep quality levels by sex and REBT sessions.

| Characteristics | Female | | Male | |
|---|---|---|---|---|
| | N | % | N | % |
| Anxiety Session 1 | | | | |
| Normal | 0 | 0 | 0 | 0 |
| Mild | 11 | 50 | 11 | 50 |
| Moderate | 6 | 54.55 | 5 | 45.45 |
| Severe | 18 | 66.67 | 9 | 33.33 |
| Anxiety Session 3 | | | | |
| Normal | 5 | 71.43 | 2 | 28.57 |
| Mild | 12 | 46.15 | 14 | 53.85 |
| Moderate | 18 | 66.67 | 9 | 33.33 |
| Severe | 0 | 0 | 0 | 0 |
| Very Severe | 0 | 0 | 0 | 0 |
| Anxiety Session 6 | | | | |
| Normal | 17 | 60.71 | 11 | 39.29 |
| Mild | 16 | 57.14 | 12 | 42.86 |
| Moderate | 2 | 50 | 2 | 50 |
| Depression Session 1 | | | | |
| Normal | 1 | 100 | 0 | 2.86 |
| Mild | 13 | 54.17 | 11 | 45.83 |
| Moderate | 4 | 80 | 1 | 20 |
| Severe | 10 | 76.92 | 3 | 23.08 |
| Depression Session 3 | | | | |
| Normal | 9 | 56.25 | 7 | 43.75 |
| Mild | 8 | 61.54 | 5 | 38.46 |
| Moderate | 11 | 73.33 | 4 | 26.67 |
| Severe | 7 | 46.67 | 8 | 53.33 |
| Very Severe | 0 | 0 | 1 | 100 |
| Depression Session 6 | | | | |
| Normal | 18 | 62.07 | 11 | 37.93 |
| Mild | 13 | 68.42 | 6 | 31.58 |
| Moderate | 4 | 40 | 6 | 60 |
| Severe | 0 | 0 | 2 | 100 |
| Sleep Quality Session 1 | | | | |
| Good | 4 | 50 | 4 | 50 |
| Poor | 31 | 59.62 | 21 | 40.38 |
| Sleep Quality Session 3 | | | | |
| Good | 21 | 47.73 | 23 | 52.27 |
| Poor | 14 | 87.50 | 2 | 12.50 |
| Sleep Quality Session 6 | | | | |
| Good | 35 | 58.33 | 25 | 41.67 |

Among male participants who experienced anxiety in the first session of REBT, 50% had mild anxiety, 45.45% had moderate anxiety, and 33.33% had severe anxiety. In the third session of REBT, there were improvements in mild and moderate anxiety, while the number of participants with normal conditions (non-anxious) increased by 28.5%. By the sixth session of REBT, mild anxiety improved in 41.86% of participants, and the number of participants with normal conditions increased to 39.29%.

For male participants who reported depression in the first REBT session, mild depression was present in 45.83%, severe depression in 23.08%, and moderate depression in 20%. In the third session of REBT, there was an improvement in mild depression (38.46%). The depression continued to improve in the sixth session of REBT, with mild depression present in only 3.58% of participants, and the number of participants with normal conditions (non-depressed) increased by 37.9%. During the initial REBT session, 59.62% of the female and 40.38% of the male participants reported having poor sleep quality. However, by the sixth REBT session, all participants reported an improvement in sleep quality, with 58.33% of the female and 41.67% of the male participants reporting good sleep quality, indicating an improvement from the initial session.

### 3.2. Differences in the Proportion of Characteristics, Anxiety, Depression, and Sleep Quality with REBT Therapy Sessions by Sex

The study's findings suggest that there were no significant differences in the rates of anxiety and depression between men and women over the six REBT therapy sessions. However, there was a significant difference between men and women in terms of sleep quality during the third REBT session ($p = 0.001$), as shown in Table 2.

**Table 2.** Differences in the proportion of characteristics, anxiety, depression, and sleep quality with REBT therapy sessions by sex.

| Variables | Female | | Male | | $x^2$ | $p$ |
|---|---|---|---|---|---|---|
| | N | % | N | % | | |
| Age | | | | | | |
| <40 yo | 18 | 56.25 | 14 | 43.75 | | |
| ≥40 yo | 17 | 60.71 | 11 | 39.29 | 0.12 | 0.73 |
| Work Status | | | | | | |
| Unemployed | 11 | 52.38 | 10 | 47.62 | | |
| Private Employee | 12 | 70.59 | 5 | 29.41 | | |
| Public Employee | 5 | 71.43 | 2 | 28.57 | 2.69 | 0.44 |
| Entrepreneur | 7 | 46.67 | 8 | 53.33 | | |
| Education | | | | | | |
| Junior High School | 3 | 37.50 | 5 | 62.50 | | |
| High School and Vocational Education | 14 | 53.85 | 12 | 46.15 | | |
| Higher Education (Diploma and Bachelor's) | 18 | 69.23 | 8 | 30.77 | 2.91 | 0.23 |
| Marital Status | | | | | | |
| Married | 22 | 64.71 | 12 | 35.29 | | |
| Unmarried | 13 | 50 | 13 | 50 | 1.31 | 0.25 |
| Anxiety Session 1 | | | | | | |

**Table 2.** *Cont.*

| Variables | Female | | Male | | $x^2$ | $p$ |
|---|---|---|---|---|---|---|
| | **N** | **%** | **N** | **%** | | |
| Normal | 0 | 0 | 0 | 0 | | |
| Mild | 11 | 50 | 11 | 50 | | |
| Moderate | 6 | 54.55 | 5 | 45.45 | 1.46 | 0.48 |
| Severe | 18 | 66.67 | 9 | 33.33 | | |
| Anxiety Session 3 | | | | | | |
| Normal | 5 | 71.43 | 2 | 28.57 | | |
| Mild | 12 | 46.15 | 14 | 53.85 | | |
| Moderate | 18 | 66.67 | 9 | 33.33 | 2.85 | 0.24 |
| Severe | 0 | 0 | 0 | 0 | | |
| Very Severe | 0 | 0 | 0 | 0 | | |
| Anxiety Session 6 | | | | | | |
| Normal | 17 | 60.71 | 11 | 39.29 | | |
| Mild | 16 | 57.14 | 12 | 42.86 | 0.20 | 0.91 |
| Moderate | 2 | 50 | 2 | 50 | | |
| Depression Session 1 | | | | | | |
| Normal | 1 | 100 | 0 | 2.86 | | |
| Mild | 13 | 54.17 | 11 | 45.83 | | |
| Moderate | 4 | 80 | 1 | 20 | 5.76 | 0.22 |
| Severe | 10 | 76.92 | 3 | 23.08 | | |
| Depression Session 3 | | | | | | |
| Normal | 9 | 56.25 | 7 | 43.75 | | |
| Mild | 8 | 61.54 | 5 | 38.46 | | |
| Moderate | 11 | 73.33 | 4 | 26.67 | | |
| Severe | 7 | 46.67 | 8 | 53.33 | 3.71 | 0.45 |
| Very Severe | 0 | 0 | 1 | 100 | | |
| Depression Session 6 | | | | | | |
| Normal | 18 | 62.07 | 11 | 37.93 | | |
| Mild | 13 | 68.42 | 6 | 31.58 | | |
| Moderate | 4 | 40 | 6 | 60 | 5.14 | 0.16 |
| Severe | 0 | 0 | 2 | 100 | | |
| Sleep Quality Session 1 | | | | | | |
| Good | 4 | 50 | 4 | 50 | | |
| Poor | 31 | 59.62 | 21 | 40.38 | 0.26 | 0.61 |
| Sleep Quality Session 3 | | | | | | |
| Good | 21 | 47.73 | 23 | 52.27 | | |
| Poor | 14 | 87.50 | 2 | 12.50 | 7.64 | 0.01 * |
| Sleep Quality Session 6 | | | | | | |
| Good | 35 | 58.33 | 25 | 41.67 | | |

* $p < 0.05$ = significant.

*3.3. Repeated Measures ANOVA Analysis*

The result showed significant differences in anxiety, depression, and sleep quality scores during the first, third, and sixth REBT sessions at the Atma Jaya Hospital's outpatient psychiatric polyclinic, as shown in Table 3 and Figure 1.

**Table 3.** Analysis of repeated measures ANOVA.

| | REBT Session 1 | | REBT Session 3 | | REBT Session 6 | | $x^2$(df) | *p* | Kendall's W |
|---|---|---|---|---|---|---|---|---|---|
| | Median | Min–Max | Median | Min–Max | Median | Min–Max | | | |
| Anxiety | 12.50 | 6–19 | 9 | 3–14 | 5.5 | 2–10 | 118.56 (2) | <0.001 * | 0.99 |
| Depression | 14.50 | 4–23 | 10 | 3–21 | 6.5 | 3–19 | 108.18 (2) | <0.001 * | 0.90 |
| Sleep Quality | 9 | 2–16 | 5 | 3–14 | 4 | 2–5 | 75.95 (2) | <0.001 * | 0.63 |

* *p* < 0.001 = significant.

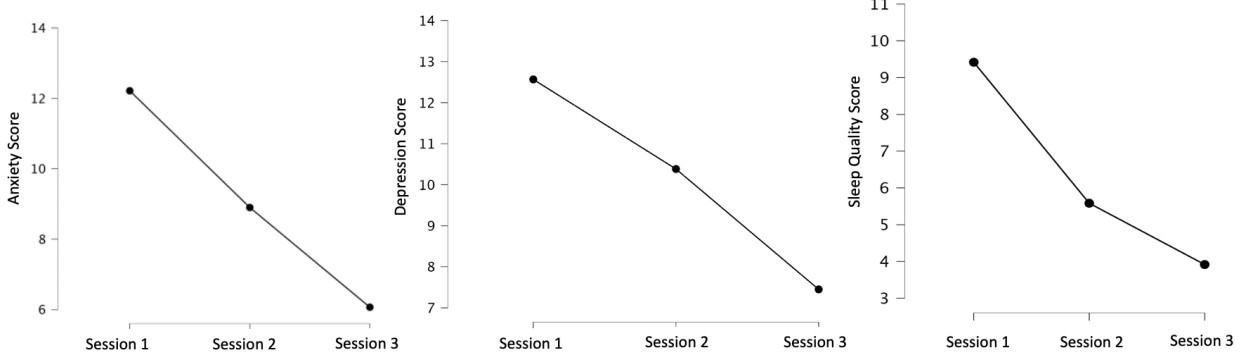

**Figure 1.** Anxiety, depression, and sleep quality improvement scores with REBT sessions.

## 4. Discussion

The COVID-19 pandemic brought about multiple stressors in people's lives, including the fear of contracting the virus, losing loved ones, experiencing job loss, and decreasing income [3]. The media's constant reporting of COVID-19 cases, illnesses, and deaths added to this fear and stress. As a result, those who previously did not experience anxiety and those who did became excessively anxious during the pandemic [19]. Online surveys conducted in 5 Indonesian provinces with the highest rates of COVID-19 cases in Indonesia showed that 64.3% of 1522 participants had psychological problems such as anxiety and depression. Symptoms of anxiety and depression included excessive fear and worry, difficulty relaxing, sleep disturbances, and heightened alertness [3].

In line with our research purpose, REBT is a psychotherapy approach that emphasizes the role of beliefs in shaping emotions and behaviors. It suggests that people's reactions to events are not directly caused by those events, but rather by their beliefs about them. REBT distinguishes between rational and irrational beliefs and proposes that individuals can respond to adversity in either healthy or unhealthy ways. Studies have found that irrational beliefs are associated with negative emotions, various mental health conditions, and maladaptive behaviors. To address this, REBT focuses on reducing irrational beliefs and promoting rational ones. Our research approach can be useful in psychotherapy for improving mental health outcomes through public and private outpatient settings [20]. In this vein, the demographics of participants in the study aligned with the outpatient data from Atma Jaya Hospital mental health clinics in 2018–2020, which showed that 55% of the patients were women between the ages of 15–84; 48% of BPJS (public) patients had a junior high school education level, and approximately 39% of Bonaventura (private) patients had a D3/S1 (diploma/bachelor's) education level.

Although not statistically significant, our study found that women were more likely to experience anxiety, depression, and poor sleep quality compared to men. The study found that at the beginning of REBT sessions, female participants had varying degrees

of anxiety, including severe (50%), moderate (54.55%), and mild anxiety (50%). However, in session 3 REBT, there were improvements in moderate (66.67%) and mild (46.15%) anxiety. By session 6 REBT, participants reported mild (57.14%) and moderate (50%) anxiety. Similarly, at the beginning of REBT sessions, female participants with depression had mild (54.17%), moderate (80%), and severe (76.92%) depression. In session 3 REBT, there were improvements in the rates of moderate (73.3%), mild (61.54%), and severe (46.67%) depression. Poor sleep quality was prevalent among both female (59.62%) and male (40.38%) participants at the beginning of REBT. However, by session 6 of REBT, there was an improvement, with all female (58.33%) and male (41.67%) participants reporting good sleep quality.

In line with this reality, another survey conducted by Srifianti showed that 57% of people with anxiety were women, while 53% were men. Similarly, 61% of depressed participants were women, and 55% were men. Women aged 20–30 years with junior high school education and who were not working were found to be the most anxious and depressed [21]. According to Jones and Nelson (2011), negative thoughts and emotions are closely related, leading to negative self-talk and irrational beliefs that impact behavior and emotions [22]. People with anxiety and negative thoughts often have negative views about various problems they face in their lives [23]. To overcome these irrational beliefs, it is necessary to use a practical cognitive approach such as REBT, which emphasizes how our way of thinking can influence our feelings (emotions), shaping the individual's behavior. The emergence of irrational thoughts later in REBT therapy will be converted into rational thoughts so that clients can live their lives and engage in self-development.

Similarly, Abate, K.H. has stated that individuals with depression often have negative views about life, coming from cognitive schemes adopted as children based on early learning experiences. For these individuals, even small failures are raised out of proportion, signifying the experience of defeat. The symptoms include loss of pleasure, guilt, loss of interest, and changes in sleep patterns [24]. Szentagotai stated that depressed participants on REBT were helped to recognize negative thoughts and turn them into positive ones based on constructivism principles; everyone creates their reality. Therapists and individuals collaborated to identify Socratic dialogue in recognition of the existence of dysfunctional beliefs and automatic thoughts. Next, daily activity plans were made to improve the circle of problems, starting from planning daily activities [7]. Lipsky, M.J.; Kassinove, H.; and Miller, N.J. have stated that individuals with depression are less likely to be engaged in personal activities that create pleasure [25]. During the REBT sessions, these individuals recognized positive feelings through the mastery of small things to help them feel better. Finally, after REBT therapy, the subjects gradually experienced changes in their behavior for the better [25,26].

The sex/gender-related differences in sleep quality might be related to depression, anxiety, and sleep disorders, such as insomnia [5]. Research shows that the prevalence of poor sleep quality in women is 7% higher than in men. Depression and anxiety disorders are interchangeably linked to poor sleep quality [27]; one study found that 26% of participants experienced depression, while 65.7% had poor sleep quality [28]. Research showed that 42% of participants experienced anxiety, and around 59% had poor sleep quality [29]. The bi-directional relationship between depression/anxiety and poor sleep quality existed before the COVID-19 pandemic, and various regulations such as large-scale social restrictions (PSBB), work from home (WFH), and distance learning (PJJ) [29–31] exacerbated this relationship.

In this study, participants underwent six REBT sessions. The research participants followed the therapy process and carried out their duties well. The participants understood that they were experiencing negative thoughts and problems which influenced them. With REBT, they were able to confront, attack, defend, and discuss their irrational beliefs, showing the participants more subjectively rational and optimistic patterns of healthier thoughts. Homework given in therapy sessions helped participants to practice more positive and rational ways of life and to develop new attitudes and behaviors [7,32]. This

can be seen from the results, which showed a significant decrease in anxiety, depression, and sleep quality scores after the REBT sessions ($p < 0.001$).

In line with these results, the REBT approach was effective in treating mental/emotional disorders and behavioral problems. Several studies have shown improvements in symptoms and behavior after receiving REBT intervention, emphasizing the importance of addressing thinking patterns and beliefs. Research on 68 adolescents with anxiety disorders who received ten sessions of REBT showed that patients who received REBT and pharmacotherapy had more pronounced improvements than patients who received relaxation therapy and pharmacotherapy. Patients experienced improvements in their anxiety symptoms after the tenth session of treatment [33]. In addition, a study of 170 depressive disorder patients who received eight weeks of REBT intervention showed improvements in their symptoms and decreases in Hamilton Rating Scale for Depression (HDRS) scores after six sessions (weeks) of therapy [34]. Research on 123 adolescents with behavioral disorders who received REBT intervention for 10 weeks showed improvements in their academic achievement and behavior after attending 5 sessions (5 weeks) of REBT therapy [35]. In Indonesia, REBT has begun to be used widely for treating mental/emotional disorders (depression, anxiety) and behavioral problems. A further approach would require cross-division collaboration to integrate REBT-based psychotherapy into Indonesian health services. In other research, there is an urgent need for enhanced public mental healthcare services, including surge capacity and expanded resourcing and staffing in the hospital, acute, and community sectors. Private mental healthcare services have provided care, and must be partnered with public services to enhance surge capacity. Thus, the possibility of exploring the differences between, barriers to, and opportunities in both public and private mental health settings should be explored in future initiatives [36].

## 5. Limitations

As a quasi-experimental study, lower internal validity was expected due to the non-randomized nature. However, due to our participants' data collection, we gathered demographic data that mirrored the data of the Atma Jaya Hospital's outpatient data. As a teaching hospital consisting of public and private outpatient units, it can be seen how the results are relevant to the Indonesian mental health context. Moreover, participants with lower socioeconomic backgrounds may have faced challenges with transportation costs, which could affect their ability to attend therapy sessions. To address this, we covered the transportation costs for participants, resulting in dropout rates of zero. Our study did not specifically measure the effect of REBT intervention based on sex. However, based on other studies on psychotherapy, it is known that gender does not fully moderate the therapeutic outcome [37,38]. In future studies, controlling confounding variables such as additional treatments and social determinants of health could be considered.

## 6. Summary

This research was conducted on 60 participants at the public and private mental health outpatient centers of Atma Jaya Hospital, Jakarta. The six sessions of REBT-based psychotherapy intervention significantly improved the anxiety, depression, and sleep quality of participants. In session 1 of REBT, poor sleep quality was found in females (59.62%) and 40.38% of male respondents. In session 6 of REBT, there was an improvement in sleep quality; 58.33% of female and 41.67% of male respondents had good sleep quality. Repeated measures ANOVA analysis showed significant differences ($p < 0.001$) in anxiety, depression, and sleep quality scores after 1, 3, and 6 REBT sessions. Our findings suggest that REBT-based psychotherapy is an effective intervention that can be delivered in both public and private outpatient settings to more effectively reduce the course of mental illnesses. Furthermore, this study highlights the importance of psychotherapy and cross-division collaboration in the Indonesian healthcare system.



**Author Contributions:** Conceptualization, S., A.W., M.V. and A.T.; methodology, S., A.W., M.V. and A.T.; software, A.T.; validation, S.; formal analysis, S.and A.W.; investigation, S., A.W., M.V. and A.T.; resources, S.; data curation, S. and A.T.; writing—original draft preparation, S., M.V. and A.T.; writing—review and editing, A.W.; visualization, A.W. and A.T.; supervision, S.; project administration, M.V. and A.T. All authors have read and agreed to the published version of the manuscript.

**Funding:** This research received no external funding.

**Institutional Review Board Statement:** This research was approved by the Ethical Clearance Committees, Atma Jaya Catholic University of Indonesia, No. 10/04/KEP-FKIKUAJ/2021, April 2021.

**Informed Consent Statement:** Informed consent was obtained from all subjects involved in the study.

**Data Availability Statement:** Data is unavailable due to privacy and ethical restrictions.

**Acknowledgments:** The authors thank Astri Parawita Ayu, who assisted in the data analysis of this research; the participants joining this research; and the Atma Jaya Hospital team.

**Conflicts of Interest:** The authors declare no conflict of interest.

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
