# Peer review of "REBT Integration in Public and Private Outpatient Settings during COVID-19 Pandemic: Perspectives from Indonesia"

_2673-5318, doi:10.3390/psychiatryint4020011_

Round 1

Reviewer 1 Report

I would like to thank you for the opportunity since I feel very fortunate to be able to review this article and I would like to congratulate the authors for this work. For me, this topic is very important and has a lot of value. I detail my suggestions below and at the end my consideration.

This manuscript aimed to obtain an overview of psychological conditions, especially anxiety, depression, and sleep quality, in public and private outpatient settings and to determine the future role of TREC-based interventions in Indonesia.

Title: The title is concrete, representative and indicative of the problem investigated in the manuscript. As a suggestion, the title should provide information of the subject group and would not include the acronyms TREC and CLP since a reader unfamiliar with these concepts would not understand the title.

Abstract: The abstract is clear and complies with the general rules for writing a good abstract. Before indicating the acronyms TREC, REBT... I would explain what they mean and then in parentheses indicate the acronyms. However, I would like to see a better description of the sample, indicating the context.  This is the most important section of the document since it will be read many more times than even the manuscript itself, so it needs the most attention. A brief note on the importance of the research is an excellent ending to a high-level abstract.

Introduction

As I mentioned, I find this research extremely important in contributing to the of health. I do not disagree with the authors' justifications and read many very good and current arguments.  

It is suggested to the authors that based on the stated objective they highlight the research questions that will help drive the research and discussion based on the findings found in which the study variables, study population, and expected outcome appear.

Material and method.

Instruments: Were they multidimensional instruments, was the behavior of the instruments tested in the sample taken, was any confirmatory factor analysis performed, and if so, how? The number of items and dimensions of some instruments is not indicated.

Participants. This section should be better defined. In this section (participants), the characteristics of the sample should be included.

Statistical analysis: Was the distribution of the data explored using some type of statistical test to assess whether the assumption of normality was met? What significance value was established for p? How the data for categorical and continuous variables are presented should be explained?

Results: 

The results are correctly displayed and are easy to read and straightforward for a scholar unaccustomed to quantitative methodology.

Discussion: It seems to me that a great job has been done in comparing the findings with other studies. Congratulations. It is suggested to include a section on practical and theoretical implications to evaluate the scope of the research.

Conclusions: They are clear and provide an answer to the stated objectives.

Author Response

Dear Reviewer,

I want to express our gratitude for your support in reviewing our article. We hope this publication will illuminate Indonesia's mental health research and delivery.

We have reviewed your comments thoughtfully. As a result, we edited the entire document to ensure it was easy to read and addressed all of your comments.

Title: The title is concrete, representative and indicative of the problem investigated in the manuscript. As a suggestion, the title should provide information of the subject group and would not include the acronyms TREC and CLP since a reader unfamiliar with these concepts would not understand the title.

We changed the title to “REBT Integration in Public and Private Outpatient Settings During COVID-19 Pandemic: Perspectives from Indonesia”

We changed the CLP title and kept the REBT abbreviation in the title but explained it in the abstract section.

Abstract: The abstract is clear and complies with the general rules for writing a good abstract. Before indicating the acronyms TREC, REBT... I would explain what they mean and then in parentheses indicate the acronyms. However, I would like to see a better description of the sample, indicating the context.  This is the most important section of the document since it will be read many more times than even the manuscript itself, so it needs the most attention. A brief note on the importance of the research is an excellent ending to a high-level abstract.

Due to the word limit, the REBT abbreviation is written, but leaving general info PHQ-9, GAD-7, and PSQI. The Abstract is paraphrased fully for clarity. Edited to:

This quasi-experimental study aimed to investigate how Rational Emotive Behavior Therapy (REBT) works in Indonesian outpatient settings, especially in the context of the COVID-19 pandemic. This study evaluated depression, anxiety, and sleep among 60 Atma Jaya Hospital patients (public and private) using several assessment tools (PHQ-9, GAD-7, and PSQI). The majority of participants were under 40 (53.3%), female (58.3%), married (56.6%), and working formally (65%), with either high school or diploma/bachelor level education (86.6%). After six REBT therapy sessions, significant improvements showed in anxiety, depression, and sleep quality scores (p<0.001). Participants showed improvements in anxiety and depression in the third and sixth sessions, and the sixth session for sleep quality. These findings suggest that REBT-based interventions are effective in public and private outpatient settings, highlighting the importance of psychotherapy and cross-division collaboration in the Indonesian healthcare system.

Introduction: As I mentioned, I find this research extremely important in contributing to the of health. I do not disagree with the authors' justifications and read many very good and current arguments. It is suggested to the authors that based on the stated objective they highlight the research questions that will help drive the research and discussion based on the findings found in which the study variables, study population, and expected outcome appear.

Paraphrased the introduction and highlighted the research purpose:

In 2019, the World Health Organization declared COVID-19 a global pandemic, with more than 6 million deaths globally. The WHO recommends various strategies to reduce transmission, including hand washing, masks, and physical distancing (WHO, 2021). The Special Capital Region of Jakarta has implemented large-scale social restrictions (PSBB) as one of the strategies to reduce the spread of SARS-CoV2 (Corona Jakarta, 2021). In addition, the "new normal" era was introduced to help people return to their everyday lives with various adjustments to prevent the spread of COVID-19 (WHO, 2021; Corona Jakarta, 2021).

The negative impact of a global pandemic can further harm the mental health aspects of people with or without mental disorders. With the widespread fear and uncertainty surrounding the virus, people struggle with anxiety, depression, sleep disorders, and stress during the pandemic (PDSKJI, 2021). The COVID-19 pandemic has led to many irrational thoughts and beliefs, such as the fear of contracting the virus, uncertainty of its impact, and worry about the future. The overall prevalence of depression, anxiety, distress, and insomnia was 31.4%, 31.9%, 41.1%, and 37.9%, respectively (Wu et al., 2021). This was highlighted in other research, with anxiety (67%) being the major psychological impact on the general population (Chandra et al., 2020), and how self-isolation can increase depressive symptoms (74.3%) and poor sleep quality (43.0%) (Puthran et al., 2016).

The general objective of REBT intervention is to increase the ability to manage irrational beliefs to be rational in dealing with various situations, events, experiences, or problems while increasing logical thinking, self-acceptance, sense of comfort, security, and positive behavior (Szentagotai, 2006). REBT can help individuals cope with fears by challenging and changing their irrational beliefs and replacing them with rational, more adaptive ones. Changing thinking patterns and beliefs can significantly change feelings and behavior. A recent study found that REBT was effective in reducing symptoms of anxiety and depression in individuals who were struggling with the impact of COVID-19. The study by Bechara recruited 120 participants experiencing high levels of anxiety and depression due to the pandemic. The participants were randomly assigned to REBT intervention or control group. After 12 weeks of treatment, the REBT group showed significantly reduced anxiety and depression symptoms compared to the control group (Bechara et al., 2021).

Given the challenging situation with COVID-19, this quasi-experimental study was designed to evaluate the psychological conditions of patients coming to the BPJS (Public) and Bonaventura (Private) outpatient settings at the Atma Jaya Hospital and provided REBT-based psychological interventions. Each individual will get six therapy sessions, each lasting 30-45 minutes. We aimed to obtain an overview of psychological conditions, especially anxiety, depression, and sleep quality during the COVID-19 pandemic and to determine the future role of REBT-based interventions in Indonesia. The results of this study are expected to create further REBT-based interventions and modules to overcome anxiety, depression, and sleep disorders that can be used to overcome psychological problems for patients and students in hospital and university settings.

Material and method

Instruments: Were they multidimensional instruments, was the behavior of the instruments tested in the sample taken, was any confirmatory factor analysis performed, and if so, how? The number of items and dimensions of some instruments is not indicated.

We re-detailed the design, participant, instrument, and analysis information. All measuring instruments are available and validated in Indonesian settings. Note:

The characteristics of all participants were evaluated based on age, sex, education, marital status, and occupation. To assess depression, the study used the Indonesian version of Patient Health Questionnaire (PHQ-9) that has been validated with a Cronbach's alpha value of 0.718 (Dian, 2020). For anxiety, the study used the Generalized Anxiety Disorder Scale (GAD-7), with Cronbach's alpha of 0.876 for the Indonesian version, 100% sensitivity, and 84.4% specificity (Sukmawati & Putra, 2019). Additionally, this study used the Pittsburgh Sleep Quality Index (PSQI) questionnaire, which consists of nine questions, to measure sleep quality. The Indonesian version of the PSQI questionnaire was validated in 2018 with a Cronbach's alpha value of 0.81 (Alim, 2015).

This research utilized a quasi-experimental approach and implemented six REBT sessions as the intervention. Mental health conditions of all participants were evaluated before and after each session. The REBT psychotherapy module was available, adapted, and applicable for use in the Indonesian setting (Surilena et al., 2014).

Statistical analysis: Was the distribution of the data explored using some type of statistical test to assess whether the assumption of normality was met? What significance value was established for p? How the data for categorical and continuous variables are presented should be explained?

Descriptive analysis was conducted to investigate the proportion of participants experiencing anxiety, depression, and sleep quality disorders, based on their demographic characteristics. Furthermore, the study utilized repeated measures MANOVA to evaluate changes in symptoms of anxiety, depression, and sleep quality among participants in the first, third, and sixth sessions of the REBT-based intervention. No extreme outliers were found.

Results: The results are correctly displayed and are easy to read and straightforward for a scholar unaccustomed to quantitative methodology. Discussion: It seems to me that a great job has been done in comparing the findings with other studies. Congratulations. It is suggested to include a section on practical and theoretical implications to evaluate the scope of the research.

We paraphrased the result and discussion paragraphs to make them structured and easy to read (See the attached document).

Conclusions: They are clear and provide an answer to the stated objectives.

We sufficiently summarize the conclusion, in line with all the previous sections and discussions (See the attached document).

Lastly, we are looking forward to hearing good news about our publication,

Thank you for your support!

Reviewer 2 Report

The paper presents the results from a study conducted in public and private outpatient clinics at Atma Jaya Hospital which aims to evaluate the effectiveness of REBT-based psychological interventions on the psychological condition of patients.The study includes 120 participants with high levels of anxiety and depression due to the pandemic. The participants were randomly assigned in an REBT group or a control group. The study is very well designed in terms of recruitment, data collection and data analysis. The results show that after treatment, the REBT group showed significant reductions in anxiety and depression symptoms compared to the control group. The main conclusion is that REBT-based interventions are effective psychotherapy and they can be used for reduction of severity of mental disorders.

The paper clearly and convincingly presents the results from this study and I have just a few suggestions. In the Introductory part of the paper the authors can describe in some more details the Rational Emotive Behavior Therapy (REBT) and the manner in which it conceptualizes the interdependence between thinking, emotions and behaviours with a focus of its effectiveness during the COVID-19 pandemic.

The results show also that in the beginning of the study female participants had higher anxiety, depression, and poorer sleep quality than male participants. It is interesting to comment also if the reduction of these symptoms was stronger among women who received REBT than men and what factors may explain this difference.

Author Response

Dear Reviewer,

I want to express our gratitude for your support in reviewing our article. We hope this publication will illuminate Indonesia's mental health research and delivery.

We have reviewed your comments thoughtfully. As a result, we edited the entire document to ensure it was easy to read and addressed all of your comments.

In the Introductory part of the paper the authors can describe in some more details the Rational Emotive Behavior Therapy (REBT) and the manner in which it conceptualizes the interdependence between thinking, emotions and behaviours with a focus of its effectiveness during the COVID-19 pandemic.

We paraphrased the introduction and highlighted the research purpose: Explained the context of global pandemic, psychological impact of COVID-19 pandemic, REBT intervention and its effectiveness in COVID-19 pandemic.

In 2019, the World Health Organization declared COVID-19 a global pandemic, with more than 6 million deaths globally. The WHO recommends various strategies to reduce transmission, including hand washing, masks, and physical distancing (WHO, 2021). The Special Capital Region of Jakarta has implemented large-scale social restrictions (PSBB) as one of the strategies to reduce the spread of SARS-CoV2 (Corona Jakarta, 2021). In addition, the "new normal" era was introduced to help people return to their everyday lives with various adjustments to prevent the spread of COVID-19 (WHO, 2021; Corona Jakarta, 2021).

The negative impact of a global pandemic can further harm the mental health aspects of people with or without mental disorders. With the widespread fear and uncertainty surrounding the virus, people struggle with anxiety, depression, sleep disorders, and stress during the pandemic (PDSKJI, 2021). The COVID-19 pandemic has led to many irrational thoughts and beliefs, such as the fear of contracting the virus, uncertainty of its impact, and worry about the future. The overall prevalence of depression, anxiety, distress, and insomnia was 31.4%, 31.9%, 41.1%, and 37.9%, respectively (Wu et al., 2021). This was highlighted in other research, with anxiety (67%) being the major psychological impact on the general population (Chandra et al., 2020), and how self-isolation can increase depressive symptoms (74.3%) and poor sleep quality (43.0%) (Puthran et al., 2016).

The general objective of REBT intervention is to increase the ability to manage irrational beliefs to be rational in dealing with various situations, events, experiences, or problems while increasing logical thinking, self-acceptance, sense of comfort, security, and positive behavior (Szentagotai, 2006). REBT can help individuals cope with fears by challenging and changing their irrational beliefs and replacing them with rational, more adaptive ones. Changing thinking patterns and beliefs can significantly change feelings and behavior. A recent study found that REBT was effective in reducing symptoms of anxiety and depression in individuals who were struggling with the impact of COVID-19. The study by Bechara recruited 120 participants experiencing high levels of anxiety and depression due to the pandemic. The participants were randomly assigned to REBT intervention or control group. After 12 weeks of treatment, the REBT group showed significantly reduced anxiety and depression symptoms compared to the control group (Bechara et al., 2021).

Given the challenging situation with COVID-19, this quasi-experimental study was designed to evaluate the psychological conditions of patients coming to the BPJS (Public) and Bonaventura (Private) outpatient settings at the Atma Jaya Hospital and provided REBT-based psychological interventions. Each individual will get six therapy sessions, each lasting 30-45 minutes. We aimed to obtain an overview of psychological conditions, especially anxiety, depression, and sleep quality during the COVID-19 pandemic and to determine the future role of REBT-based interventions in Indonesia. The results of this study are expected to create further REBT-based interventions and modules to overcome anxiety, depression, and sleep disorders that can be used to overcome psychological problems for patients and students in hospital and university settings.

The results show also that in the beginning of the study female participants had higher anxiety, depression, and poorer sleep quality than male participants. It is interesting to comment also if the reduction of these symptoms was stronger among women who received REBT than men and what factors may explain this difference.

Explained in the discussion and study limitations

In line with the results, REBT approach has shown to be effective in treating mental-emotional disorders and behavioral problems (p < 0.001). However, our study did not specifically measure the effect of REBT intervention based on sex. However, based on other studies on psychotherapy, gender does not fully moderate the therapeutic outcome (Cuijpers et al., 2014; Wade et al., 2016). In future studies, controlling confounding variables such as additional treatments and social determinants of health could be considered.

Lastly, we are looking forward to hearing good news about our publication,

Thank you for your support!
